# Assessing the NLRP3 Inflammasome Activating Potential of a Large Panel of Micro- and Nanoplastics in THP-1 Cells

**DOI:** 10.3390/biom12081095

**Published:** 2022-08-09

**Authors:** Mathias Busch, Gerrit Bredeck, Friedrich Waag, Khosrow Rahimi, Haribaskar Ramachandran, Tobias Bessel, Stephan Barcikowski, Andreas Herrmann, Andrea Rossi, Roel P. F. Schins

**Affiliations:** 1IUF–Leibniz-Research Institute for Environmental Medicine, 40225 Duesseldorf, Germany; 2Technical Chemistry I, Center for Nanointegration Duisburg-Essen (CENIDE), University of Duisburg-Essen, 45141 Essen, Germany; 3DWI–Leibniz Institute for Interactive Materials, Institute of Technical and Macromolecular Chemistry, RWTH Aachen University, 52074 Aachen, Germany

**Keywords:** carbon nanotubes, titanium dioxide, polymers, IL-1beta, IL-8, fibers, screening

## Abstract

Due to the ubiquity of environmental micro- and nanoplastics (MNPs), inhalation and ingestion by humans is very likely, but human health effects remain largely unknown. The NLRP3 inflammasome is a key player of the innate immune system and is involved in responses towards foreign particulate matter and the development of chronic intestinal and respiratory inflammatory diseases. We established *NLRP3*-proficient and -deficient THP-1 cells as an alternative in vitro screening tool to assess the potential of MNPs to activate the NLRP3 inflammasome. By investigating cytokine release (IL-1β and IL-8) and cytotoxicity after treatment with engineered nanomaterials, this in vitro approach was compared to earlier published ex vivo murine bone marrow-derived macrophages and in vivo data. This approach showed a strong correlation with previously published data, verifying that THP-1 cells are a suitable model to investigate NLRP3 inflammasome activation. We then investigated the proinflammatory potential of eight MNPs of different size, shape, and chemical composition. Only amine-modified polystyrene (PS-NH_2_) acted as a direct NLRP3 activator. However, polyethylene terephthalate (PET), polyacrylonitrile (PAN), and nylon (PA6) induced a significant increase in IL-8 release in *NLRP3^−/−^* cells. Our results suggest that most MNPs are not direct activators of the NLRP3 inflammasome, but specific MNP types might still possess pro-inflammatory potential via other pathways.

## 1. Introduction

The increasing production and use of plastic products since the 1950s, in combination with waste mismanagement, has led to global environmental pollution. Physical forces, UV radiation, and heat cause the embrittlement and abrasion of plastic products and waste, ultimately leading to fragments in the micro- and nanometer range [1,2]. Further sources include plastic beads in cosmetics [3] as well as synthetic microfibers originating from clothes [4]. These highly persistent particles, fragments, beads, and fibers can be found in any place on the planet, including water [5], soil [6], air [7], plants [8], animals [9], and humans [10]. The presence of these micro- and nanoplastics (MNPs) implies human exposure via oral, inhalation, and dermal routes. Accordingly, the “One Health Concept” calls for designing research on the effects of such pollutants to achieve better public health outcomes [11]. Recent in vivo studies suggest a pro-inflammatory potential of plastic particles after ingestion [12] or inhalation [13]. 

As an important player of the innate immune system, the NOD-, LRR- and pyrin domain-containing protein 3 (NLRP3) inflammasome is involved in a variety of inflammatory responses toward external stimuli, as well as in the development of chronic inflammatory diseases [14,15]. These external stimuli include viral infections, the presence of bacterial components, and foreign particulate matter. For example, lung inflammation and fibrosis induced by crystalline silica or asbestos fibers have been shown to be NLRP3-dependent [16,17,18] and several studies have suggested a connection between intestinal inflammation and the oral uptake of particulate matter [19,20,21]. Due to their ability to actively phagocytize foreign particles, macrophages play a major role in the innate immune defense in the lung and the gastrointestinal tract following inhalation or ingestion of particles [22,23]. Depending on the particle type and their specific physicochemical properties (e.g., size, shape), contact or uptake might lead to an activation of the NLRP3 inflammasome. Upon activation, the NLRP3 inflammasome recruits procaspase-1, which leads to the formation of active caspase-1. Caspase-1 cleaves the inactive pro-interleukin (IL)-1β, resulting in the formation and secretion of mature IL-1β [24]. As an early pro-inflammatory cytokine, IL-1β is able to induce subsequent pro-inflammatory cascades [25], including the activation of transcription factor nuclear factor-kappa B (NF-κB) [26]. NF κB is capable of regulating secondary inflammatory mediators such as IL-8, an important chemoattractant during inflammation [27,28].

The release of IL-1β after particle treatment of bone marrow-derived macrophages (BMDMs) isolated from wildtype (WT) or *Nlrp3* knockout mice, has previously been assessed in our laboratory as an ex vivo approach to investigate the ability of a large panel of different titanium dioxide (TiO_2_) nanomaterials to activate the NLRP3 inflammasome [29]. In light of the 3Rs, especially to replace the use of animals with alternative in vitro methods [30], we used CRISPR/Cas technology in our present study to generate THP-1 *NLRP3^−/−^* cells and utilized the THP-1 cell line as a suitable alternative to BMDMs isolated from mice. We applied the same TiO_2_ samples to compare the THP-1 cell model against BMDM data. Furthermore, we compared the pro-inflammatory potential of two multi-walled carbon nanotubes (MWCNTs) of different aspect ratios in THP-1 cells to their outcomes in a previous in vivo study in mice [31]. Ultimately, we investigated the secretion of the NLRP3-specific cytokine IL-1β, secretion of the NLRP3-independent cytokine IL-8, and release of the cytotoxicity indicator lactate dehydrogenase (LDH) in THP-1 WT and *NLRP3^−/−^* cells after treatment with a large panel of MNPs. This principle of comparing the reaction of *NLRP3*-proficient and deficient THP 1 cells as an indicator of specific NLRP3 activation has been used in infectious disease research [32,33] and has already been described for silica particles [34]. A critical aspect of state-of-the-art in vitro testing of nanomaterials is to avoid misclassification due to assay artefacts resulting from their large specific surface areas and reactivities [35]. Therefore, prior to in vitro experiments, we tested all investigated materials (TiO_2_, MWCNTs, and MNPs) for interference with the assays we used. We investigated commercially available MNPs such as polystyrene (PS), polyvinyl chloride (PVC), and polyethylene (PE), as well as MNPs generated by novel methods such as polyethylene terephthalate (PET), polyester (PES), polyacrylonitrile (PAN), and polyamide 6 (PA6, also known as nylon). Generated MNPs were thoroughly characterized prior to in vitro application.

## 2. Materials and Methods

### 2.1. Chemicals and Test Materials

RPMI 1640 cell culture medium, 2-mercaptoethanol (ME), sodium pyruvate, and phosphate-buffered saline (PBS) were purchased from Thermo Fisher Scientific. Fetal calf serum (FCS), D-glucose, penicillin/streptomycin (P/S), phorbol 12-myristate 13-acetate (PMA), lipopolysaccharides (LPS) from E. coli O111:B4, Accutase, agarose, β-nicotinamide adenine dinucleotide sodium salt (NAD), lithium L-lactate, phenazine methosulfate (PMS), iodonitrotetrazolium chloride (INT), and bovine serum albumin (BSA) were purchased from Sigma-Aldrich/Merck. Tris base was ordered from Carl Roth (Germany). The TiO_2_ nanoparticles, carbon nanotubes, and MNP samples used in the following experiments are listed in Table 1.

### 2.2. Generation and Characterization of PET Particles

The PET particles were produced by ablating a piece of PET immersed in ultrapure water (Merck Milli-Q, 18.2 MΩ/cm) with a 1064 nm nanosecond-pulsed laser (Rofin Powerline E20). This laser synthesis method has been proven to give access to highly pure nanoparticle colloids [37], providing ideal reference materials for nanotoxicological assays [38], scalable to tens of liters per hour [39], but has rarely been studied for plastic particles [40,41]. The piece of PET, which had dimensions of 17 mm × 17 mm × 5 mm was fabricated by cutting a commercial, disposable drink bottle into pieces followed by heat-induced fusion of these pieces. For this, some pieces were filled into a quadratic mold made of steel, which was placed onto a heating plate at 300 °C. The transparent PET pieces fused to a shiny white block. This block was placed into a custom-made sealable chamber made of PTFE, which had a stirrer and was filled with water. More details on the chamber can be found elsewhere [42]. All components of the setup were washed with ethanol (analytical grade), fully assembled, and autoclaved for 20 min at 121 °C and 110 kPa (CertoClav Vac Pro). Afterwards, the setup was placed onto an xyz-stage in front of a galvanometer xy-scanner with an f-theta lens with 100 mm focal length. The distance between the focus lens and the ablation chamber was adjusted to achieve a maximum ablation rate, which was indicated by a bright plasma and a loud sound. The applied laser parameters were a laser pulse duration of 5 ns, a laser repetition rate of 10 kHz, and a laser power of 3 W leading to a laser pulse energy of 0.3 mJ. The scan speed was 2 m/s. One ablation run was 30 min and yielded 20 mg of ablated material. Five runs were applied. After each run, the produced colloid was extracted from the chamber in a clean bench (W.H. Mahl MSC-II-48) under UV-C radiation and was then collected in a flask. The setup was prepared for the next run as described above. The collected colloid was shock-frozen with liquid nitrogen and then freeze-dried (Christ Alpha 1–4) at 30 Pa.

For size analysis, 4 ml of the PET colloid was separated into the nanoparticle and microparticle fraction by sedimentation at 1 g for 10 min. Then 10 µl of the supernatant, i.e., the nanoparticle fraction, was drop casted on a carbon-coated copper grid. Analysis of the nanometer size fraction was performed by TEM (JEOL JEM 1400 Plus). The sediment, i.e., the microparticle fraction, was redispersed in 4 ml of ultrapure water and 1 ml of the suspension was drop casted on a glass slide. Analysis of the micrometer size fraction was achieved by light microscopy (Olympus CX40).

The chemical structure of the laser-generated micro- and nanoparticles was investigated by FT-IR (Shimadzu IRTracer-100). A portion (10 µg) of each fraction was dried and measured in an ATR setup. The PET target was also analyzed.

### 2.3. Production and Characterization of PES, PAN and PA6 Microfibers

Model fibrous microplastics were prepared using the method initially developed by M. Cole [43]. Briefly, three types of textile-relevant fibers, including PES and PAN (Reinhard Strauss GmbH & Co. KG, Viersen, Germany) and PA6 (Heimbach Group GmbH, Düren, Germany), were aligned, embedded in a water-soluble freezing agent (Tissue-Tek^®^ O.C.T. Compound), and sectioned into microfibers with a length of 10 μm. The sectioning was performed using a cryotome (Leica CM1950). The freezing agent was washed off and powders of microfibers were obtained using lyophilization. The resulting particles were imaged via a Hitachi S3000 scanning electron microscope (SEM) and their size distribution was analyzed using ImageJ.

### 2.4. Cell Culture

THP-1 (ATCC, TIB-202) cells were cultured in RPMI 1640-based cell culture medium (containing L-glutamine and 25 mM HEPES) substituted with 10% FCS, 1% P/S, 1 nM sodium pyruvate, 0.7% D-glucose, and 50 nM ME at 37 °C and 5% CO_2_ atmosphere. Cells were maintained at concentrations between 2 × 10^5^ and 8 × 10^5^ cells/mL.

### 2.5. Generation of an NLRP3-Deficient THP-1 Cell Line Using the CRISPR/Cas9 System

*NLRP3* THP-1 mutant cells were generated as previously described [44]. Briefly, gRNAs were designed using the CRISPR design tool CHOPCHOP (http://chopchop.cbu.uib.no/ accessed on 12 June 2019) and cloned into a modified PX458 plasmid (Addgene #48138). The resulting bicistronic vector encoded the respective gRNA, Cas9 nuclease, and a GFP selection marker. The efficiency of gRNAs was assessed using high-resolution melt analysis (HRMA). An efficient gRNA targeting *NLRP3* exon 2 (5′-GCTAATGATCGACTTCAATG-3′) was chosen for further experiments (HRMA primers: Fwd 5′-CAGACCATGTGGATCTAGCC-3′ and 5′-TGTTGATCGCAGCGAAGAT-3′). THP 1 cells were electroporated using a Neon transfection system (Thermofisher, Waltham, MA, USA) according to the manufacturer’s instructions. The transfection is only transient and all transfected elements degrade after a few days. After 48 h, cells were FACS sorted and plated as single cells into 96-well plates. Cells were duplicated into maintenance and lysis plates after a week. Clones were then lysed with proteinase K and genotyped by PCR followed by deep sequencing using a miSeq Illumina sequencer and a V2 Nano cassette. In addition to performing the sequencing which confirmed the lesion in the *NLRP3* locus, we tested the cells for the abrogation of IL-1β secretion upon activation of NLRP3 by the known NLRP3 activators PS-NH_2_ and LPS. We tested three different THP-1 *NLRP3* knockout clones and they all displayed the abrogation of IL-1β secretion (see Appendix A).

### 2.6. Experimental Procedure

For experiments, 3 × 10^6^ THP-1 cells at passages 5–10 were differentiated to a macrophage-like cell type in 25 cm^2^ flasks using 100 nM PMA. After 24 h, differentiated THP-1 cells were washed with PBS, detached with Accutase, and seeded in 24-well plates (1.2 × 10^5^ cells/well in 1 mL complete culture medium) and allowed to re-attach for 1 h. After 1 h of re-attachment, the medium was removed and cells were treated with 1 mL particle suspensions: MNPs, TiO_2_ nanoparticles, and carbon nanotubes were brought into suspension (4 mg/mL) with treatment medium (RPMI-based cell culture medium containing 1% FCS) and a 10 min sonication was performed using a Branson Sonifier 450 at a duty cycle of 0.2 sec and an output level of 5.71 (240 watts). Sonicated suspensions were diluted to obtain 50 µg/cm^2^ exposure concentration in 1 mL volume. Exposure with 10 ng/mL LPS in 1 mL medium was used as a positive control. The volume of sonicated medium was the same in all control and exposure groups. In the case of PE (density < 1 g/cm^3^), particles were embedded in the superficial layer of agarose to enable contact between buoyant particles and cells. The PE stock suspension was prepared as described previously [36] and diluted to 50 µg/cm^2^ in 1% agarose, then transferred into 24-well plates. The plates were centrifuged at 2000× *g* for 1 h at 40 °C, causing the PE particles to accumulate at the surface of the liquid agarose, and centrifuged again at 4 °C, causing the agarose to become solid. After solidification of the agarose, 1.2 × 10^5^ differentiated THP-1 cells per well were seeded on top of the particle layer. Agarose without particles was used as an additional negative control. After 24 h of treatment, supernatants were collected, the LDH assay was performed immediately, and residues were stored at −20 °C (ELISA).

### 2.7. Cytokine Quantification by Enzyme-Linked Immune-Sorbent Assay (ELISA)

The release of IL-1β and IL-8 into the supernatant following treatment was analyzed using R&D systems DuoSet ELISA kits. Antibodies were diluted according to the manufacturer’s protocol and high-protein-binding 96-well plates were incubated with capture antibody in coating buffer (0.1 M NaHCO_3_, pH 8.2) overnight. After blocking with 3% BSA/PBS, 100 µl of supernatants, diluted if necessary (IL-1β: 5 to 20-fold and IL-8: 10 to 20-fold), were incubated for 2 h. Detection antibody, horseradish peroxidase (1:40 in 1% BSA/PBS), and BioRad TMB Peroxidase EIA Substrate was consecutively incubated for 2 h, 0.5 h, and 5–20 min, respectively, before stopping the reaction with 50 µL 1 M H_2_SO_4_. Absorbance was measured at 450 and 540 nm. The standard curve was plotted as a four-parameter log fit. After 24 h incubation, cytokine concentrations were quantified via ELISA and compared to the particle-free control. Interference of all materials with the ELISA was assessed by spiking cell-free particle samples (50 µg/cm^2^) with 500 pg/mL of each cytokine. Interference of materials with the assay was not included in the calculation of the results but was considered when interpreting the data.

### 2.8. Assessment of Cytotoxicity via LDH Assay

Following 24 h of exposure, 50 µL of supernatant was transferred onto 96-well plates (in duplicate) and 150 µL of reaction mix containing 50 µL Tris buffer (200 mM, pH 8), 50 µL Li-Lactate solution (50 mM), 46 µL NAD^+^ solution (5 mM), 2 µL INT solution (65 mM), and 2 µL PMS solution (29 mM) was added per well. After 10 min incubation, the reaction was stopped with 50 µL 1 M H_2_SO_4_ and absorbance was measured at 490 and 680 nm. Interference of all materials with the LDH assay was assessed by spiking particle samples (50 µg/cm^2^) with diluted cell lysate. After 24 h incubation, LDH activity (490–680 nm) was quantified and compared to the particle-free control. Since all experiments were performed side-by-side with the same number of cells per well, with differentiated THP-1 cells that do not proliferate, no corrections for differences in absolute cell numbers were applied. The variation between different experimental runs was accounted for as described in Section 2.9.

### 2.9. Statistical Analysis

In vitro experiments were performed in at least four independent experiments, each using duplicate wells per treatment condition. For the LDH assay, the absorbance at 680 nm–490 nm was used for statistical analysis. For ELISAs, the natural logarithm of the cytokine concentration was used. In case the measured cytokine concentrations were below the limit of detection (LOD), half the LODs, i.e., 1.95 pg/mL for IL-1β and 15.63 pg/mL for IL-8, was used. To test statistical significance, mixed effect models were applied using R version 4. In the mixed effect models, exposure and genotype were applied as fixed factors. The experimental run was applied as random factor. For experiments with LPS, TiO_2_, and carbon nanotubes, *p*-values were adjusted using Dunnett’s post-hoc test. For experiments with plastics, *p*-values were adjusted using Šídák’s post hoc test because two different controls were used, i.e., without and with agarose. A *p*-value of <0.05 was considered statistically significant. The results were illustrated using GraphPad Prism version 9.1 (GraphPad Software, LLC, San Diego, CA, USA).

## 3. Results

### 3.1. PET Particles Generated by Laser Ablation

The products of PET laser ablation in water were a mixture of microparticles and nanoparticles. The nanoparticle fraction accounted for 60% of the total mass of the ablated material, as determined in a sedimentation experiment. Figure 1A shows the size distribution of the nanoparticle fraction with a mean value of 16.4 nm derived from TEM analysis. Individual particles with dimensions up to 120 nm were found. The particles exhibited different shapes ranging from nearly circular to irregular structures, which were not clearly dependent on particle size (Appendix A). The microparticle fraction mainly contained particles with diameters below 10 µm (Figure 1B). Similar morphologies compared to the nanoparticle fraction were observed (Appendix A). Larger microparticles with diameters greater than 200 µm usually had a branched or network-like morphology.

The chemical structure of both size fractions was investigated by FT-IR spectroscopy. Both fractions showed strong chemical similarity to the PET target (Appendix A). Consequently, the chemical structure of PET was largely preserved during laser ablation. However, the ablation products exhibited lower amounts of phenyl-related bonds compared to the PET target, likely due to laser-induced photodegradation of PET. The ratio of C=O to C-O ester bonds was also lower for the ablation products. In addition, a higher intensity of C-H stretching (2915 1/cm) and H-C=O groups (2800 1/cm) was observed for the nanoparticles than for the microparticles. Moreover, the spectrum of the dried nanoparticle fraction shows features of OH groups and possibly even water, indicating hydroxylation of the plastic during laser ablation in the presence of water. Note that hydroxylation also happens during the plastics’ degradation in environmental water, in accordance with what has been observed in UV-exposed plastic materials in the environment. Overall, laser ablation of PET in water yields highly pure (that is, surfactant-free) colloidal MNPs, with preserved chemical structure and higher hydroxylation for the nanoparticles compared to the microparticles.

### 3.2. Characterization of PES, PAN, and PA6 Microparticles Produced via Cryotome

SEM analysis revealed the majority of produced microfibers to be of ~10 µm length, while small quantities of longer or shorter fibers were also present in all three samples (Figure 2). Due to the low ratio of length/diameter, the majority of produced plastic particles describe a cylindric shape rather than a true fiber shape (Appendix A). However, due to limitations of the methodology, singular longer particles, exhibiting a true “fiber” shape, were also observed. Image analysis further revealed PAN fibers to possess a fissured surface on the outside as well as on the sectional areas, while the surfaces of PES and PA6 fibers appeared smooth within the resolution of the electron microscope.

### 3.3. Interference of Materials with the ELISA and LDH Assay

Prior to in vitro experiments, all investigated materials were tested for interference with the ELISA and LDH assay. None of the samples interfered with the IL-1β ELISA (Appendix A). Incubation with the TiO_2_ materials NT2 and NT3 caused a decrease of recovered IL-8 to approximately 50% (Appendix A). Furthermore, incubation with PS-NH_2_ caused a two-fold increase in detected IL-8 compared to the control. No interference with the LDH assay was observed for any of the materials (Appendix A).

### 3.4. Characterization of THP-1 Cells as a Screening Tool for NLRP3 Activation

First, THP-1 WT and *NLRP3^−/−^* cells were treated with the positive control LPS in order to confirm the successful knockout and to evaluate the differences between both cell lines regarding cytokine secretion. To interpret the cytokine data in the light of possible cytotoxic effects, the LDH assay was performed in addition to ELISA. After 24 h of treatment, the release of IL-1β and IL-8 into the supernatant was quantified, and LDH activity in the supernatant was assessed (Figure 3).

Treatment of WT cells with LPS induced a significantly increased release of IL-1β, compared to the untreated control (~800 pg/mL vs. ~60 pg/mL). IL-1β secretion by *NLRP3^−/−^* cells was either below or barely above the detection level of the ELISA, independent of LPS exposure. Furthermore, untreated WT cells exhibited an IL-8 concentration of ~1100 pg/mL in the supernatant, while the concentration released from *NLRP3^−/−^* cells was significantly lower at ~25 pg/mL. However, treatment with LPS caused a significant increase of IL-8 to ~15,000 pg/mL in both cell lines. Background LDH activity in the supernatant of both cell lines showed no difference, while treatment with LPS induced a slight increase only in the WT cells. Increased secretion of IL-1β and IL-8 was also observed following exposure of THP-1 cells to crystalline silica, and both effects were significantly stronger for WT cells than for *NLRP3^−/−^* cells (data not shown).

To compare the reaction of THP-1 cells with the reaction of BMDMs reported by Kolling et al. [29], WT and *NLRP3^−/−^* cells were treated with four different samples of TiO_2_ nanoparticles (Figure 4).

Of the four tested TiO_2_ samples, NT1, NT2, and NT4 induced a significant increase in IL-1β secretion in THP-1 WT cells. After treatment with NT2, IL-8 concentration in the supernatant of WT cells was significantly lower than in the untreated control. As described above, IL-8 release in response to NT2 and NT3 may be underestimated due to interference with the ELISA. Furthermore, no effects regarding cytokine secretion were observed in *NLRP3^−/−^* cells. Of the different TiO_2_ treatments, only NT4 induced a slight increase in LDH activity in the supernatants of both cell lines.

To further compare the applied model to published in vivo data, THP-1 WT and *NLRP3^−/−^* cells were treated with the two different MWCNTs, Mitsui 7 and NM400 (Figure 5). These MWCNTs were used in an in vivo study conducted by van Berlo et al. [31].

Treatment with Mitsui 7 induced a significant increase of IL-1β, as well as an increased activity of LDH in the supernatant, all exclusively in the WT cells. However, a significant increase of IL-8 secretion was observed in *NLRP3^−/−^* cells and a similar trend, which failed to reach statistical significance, in WT cells. Treatment with the shorter NM400 tubes caused a slight decrease in cytokine release.

### 3.5. Screening of Eight Different MNPs for NLRP3 Activation in THP-1 Cells

To finally assess the ability of MNPs to activate the inflammasome and induce a pro-inflammatory reaction, THP-1 WT and *NLRP3^−/−^* cells were treated with a large panel of different samples (Figure 6).

Within this group of plastic samples, only PS-NH_2_ induced a significant increase in IL-1β secretion in WT cells, accompanied by a strong cytotoxic effect as indicated by an increased release of LDH into the supernatant. The cytotoxic effect was also present in the *NLRP3^−/−^* cells, while no effect on IL-1β release was observed. Furthermore, the secretion of IL-8 was significantly reduced after PS-NH_2_ treatment of the WT cells. PET, PAN, and PA6 particles stimulated a significant increase of IL-8 secretion in *NLRP3^−/−^* cells, but only non-significant increases in the WT cells. The other types of plastic particles did not show any effects in the investigated endpoints.

## 4. Discussion

The present study aimed to employ an animal-free in vitro tool for investigating NLRP3 inflammasome activation in a macrophage-like cell type and to subsequently apply a large panel of different MNPs. To prove the involvement of the inflammasome in particle effects, we generated an *NLRP3*-deficient THP-1 cell line. PMA-differentiated THP-1 cells are broadly used as a model for macrophages in monocultures [45,46] or co-culture models of the intestine [47,48] and respiratory tract [49,50]. First, we confirmed the knockout using the NLRP3 activator, LPS. Next, we compared this THP-1 tool against in vitro BMDM data using TiO_2_ nanoparticles and against murine in vivo data using MWCNTs. Finally, we applied different samples of MNPs to investigate NLRP3 activation. In addition to analyzing IL-1β, an NLRP3-specific cytokine, we investigated IL-8 secretion and cytotoxic effects as unspecific, pro-inflammatory reactions to particle exposure.

After the successful knockout of the *NLRP3* gene, THP-1 cells exhibited virtually no IL-1β release, even after treatment with strong inflammasome activators like LPS [51] or PS-NH_2_ [52]. *NLRP3^−/−^* cells also showed a drastically reduced background release of IL-8, which is most likely the consequence of absent IL-1β. Even without particle exposure, THP-1 cells produce IL-1β, which acts as an activator of the NF-κB pathway [26]. In turn, NF-κB regulates IL-8 expression [27,28]. However, despite the far lower background levels of IL-8, *NLRP3^−/−^* cells show the same IL-8 levels as the WT after LPS treatment, implying that they still possess the full potential of releasing cytokines not directly depending on NLRP3 activation.

To justify the use of THP-1 WT and *NLRP3^−/−^* cells as a particle screening tool for NLRP3 activation, we compared their reaction to previously published data generated with BMDMs isolated from WT and *Nlrp3* knockout mice. In the study by Kolling et al., BMDMs were treated with the four different TiO_2_ samples (NT1-4) and IL-1β release was assessed [29]. TiO_2_ in its nanoform affects the extent of intestinal inflammation in an NLRP3-dependent manner and triggers the activation of NLRP3 in human macrophages [53]. Although the magnitude of the effects is much higher in BMDMs, presumably due to species differences, the ranking of the different TiO_2_ materials presented by Kolling et al. is in agreement with our data obtained from THP 1 cells: treatment with NT2 caused the highest increase in IL-1β release, followed by NT4 and NT1. Furthermore, our results confirm that the IL-1β release induced by TiO_2_ nanoparticles is NLRP3-dependent.

Next, we compared the reaction of THP-1 cells to MWCNT treatment to an in vivo study [31]. Long, rigid, needle-like carbon nanotubes are potent activators of the NLRP3 inflammasome, very similar to asbestos fibers [16]. Although the specific increase of IL-1β was not statistically significant, van Berlo et al. reported a much more profound pro-inflammatory and associated fibrogenic potential of the longer, rigid “Mitsui 7” MWCNTs in general, compared to the shorter, more entangled NM400. This is in strong agreement with our study, as “Mitsui 7” induced an increase of IL-1β secretion as well as low, NLRP3-dependent cytotoxic effects, which are not present with NM400. Cytotoxic properties of “Mitsui 7” were also observed by van Berlo et al. in the murine lung.

Based on the strong correlation with previously published data in other models, our THP-1 based NLRP3 activation testing assay could, upon further evaluation and validation steps, present an additional early in vitro screening approach for implementation in a hierarchical testing strategy during the hazard assessment of nanoparticles, fibers, and MNPs. Our present approach could complement the “Tier 1” in an integrated approach to testing and assessment (IATA), as recently proposed for nanomaterials regarding oral exposure [54]. However, assessment of possible interference with the assays used prior to in vitro testing is necessary to correctly interpret the obtained results [35]. For example, interference of TiO_2_ particles has been described for photometric biological assays by absorption of light or adsorption of proteins on the particle surface [55,56], as also observed for IL-8 in the present study.

Finally, we evaluated the NLRP3 inflammasome activating potential of a large panel of different MNPs. Except for PS-NH_2_ particles, which have been previously investigated in detail by Lunov et al. [52], none of the MNP samples activated the NLRP3 inflammasome in THP-1 cells, at the tested concentration. In addition to activating the NLRP3 inflammasome, causing a strong increase in IL-1β secretion, PS-NH_2_ particles show strong cytotoxicity, which may explain the significant reduction of IL-8 secretion.

There have been multiple reports of pro-inflammatory effects of MNPs in simple [57,58] and complex in vitro models [21,36], as well as in in vivo studies [12,59], but our presented data suggest that the mechanisms behind these effects are, with the exception of PS-NH_2_, NLRP3-independent. In contrast, some in vivo studies reported the activation of the NLRP3 inflammasome in heart tissue [60] or ovaries [61] of rats after 90 days of oral uptake of PS microplastics. The authors suggest an activation of NLRP3 in these tissues as a consequence of systemic inflammation and oxidative stress, instead of direct interaction of microplastics with cardiac or ovarian immune cells. Furthermore, these studies did not investigate the presence of microplastics in target tissue. Therefore, the possibly indirect activation of NLRP3 in secondary tissue after subchronic exposure is hardly comparable to the direct activation of NLRP3 in macrophages after direct single-exposure as observed in the present study.

Interestingly, PET, PAN, and PA6 particles induced a small, but statistically significant increase in IL-8 secretion in the *NLRP3^−/−^* cells, without any effect on IL-1β release. This observation points toward an NLRP3-independent pro-inflammatory effect. A background level of IL-1β, caused by a constitutively active inflammasome in PMA-differentiated THP-1 cells [62], might be responsible for masking such a weak effect in WT cells, as IL-1β acts as an activator of NF-κB [26], leading to an already high background secretion of IL-8 in WT cells. Since PET, PAN, and PA6 particles are not commercially available and the generation procedures in our study are quite novel, literature data on the toxicity of these materials are scarce. Beijer et al. reported an increase of IL-8 release in THP-1 cells after exposure to environmentally-sourced PET microparticles (20–50 µm) at a comparable exposure concentration (150 µg/mL, which translates to ~56 µg/cm^2^) [63]. Furthermore, Magrì et al. produced PET particles in a similar manner as in the present study and assessed their toxicity in a Caco-2 intestinal transwell model [40]. In contrast to the present study employing infrared laser pulses, the authors used a UV laser resulting in nanoplastics of ~100 nm diameter. No toxic effects were reported, but PET particles were internalized by the epithelial cells and showed a high propensity to cross the gut barrier. Taken together with our findings, ingested PET particles might exhibit toxicologically relevant, pro-inflammatory effects by crossing the epithelial barrier of the intestine and interacting with intestinal macrophages of the lamina propria. The PET particles used in this study are a mixture of micro- and nanoparticles. Thus, size dependency and the underlying mechanisms of the observed effect will be investigated in future studies. We want to point out that the PET particles used in this study have undergone chemical transformation during the laser ablation process, which could have an influence on their toxicological potential. However, irradiation with light and the presence of water are also factors applicable to environmentally aged plastic particles.

## 5. Conclusions

MNPs do not appear to act as potent NLRP3 inflammasome activators but can nevertheless provoke pro-inflammatory effects which remain to be mechanistically investigated. The use of THP-1 WT and *NLRP3^−/−^* cells as a screening tool for inflammasome activation presented here represents a valid non-animal alternative for the use of BMDMs isolated from WT or *Nlrp3^−/−^* mice. In view of the recognized importance of the NLRP3 pathway in particle-induced diseases, such as MWCNT-induced lung fibrosis, we propose to use our approach as an early in vitro screening tool when assessing the hazard of particles, fibers, and MNPs in a hierarchical testing strategy.

## Figures and Tables

**Figure 1 biomolecules-12-01095-f001:**
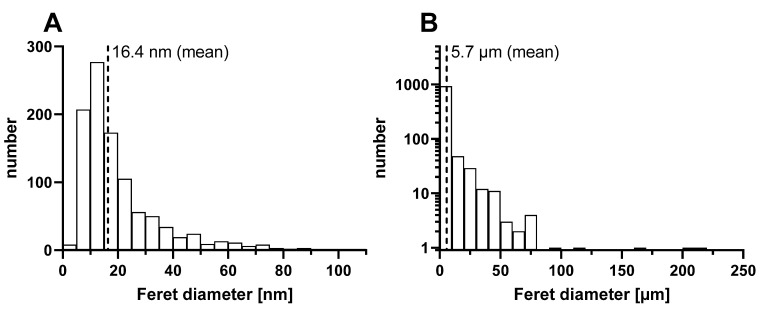
(**A**) Particle size distribution of laser-generated colloidal PET nanoparticles determined by TEM analysis (1011 particles were analyzed). (**B**) Particle size distribution of PET microparticles determined by light microscopy at a magnification of 5× (1050 particles were analyzed).

**Figure 2 biomolecules-12-01095-f002:**
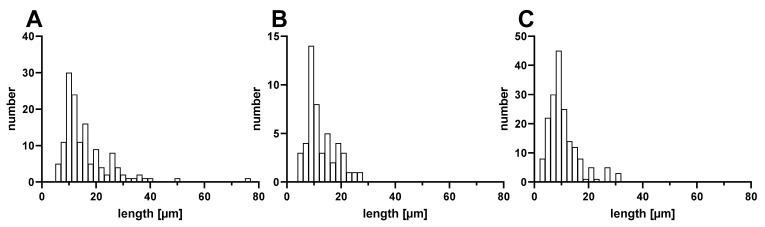
Fiber length distribution of (**A**) PES, (**B**) PAN, and (**C**) PA6 microfibers as determined by SEM analysis.

**Figure 3 biomolecules-12-01095-f003:**
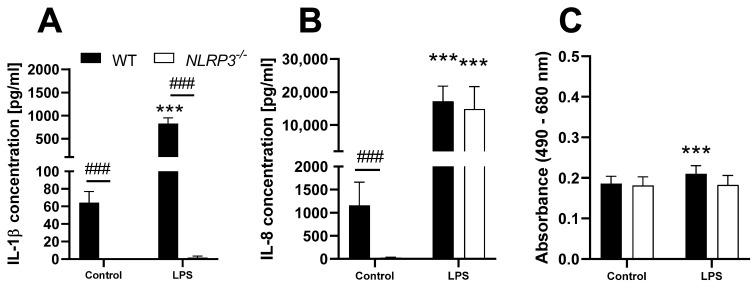
Release of IL-1β (**A**) and IL-8 (**B**), as well as LDH activity (**C**) in the supernatant of PMA-differentiated THP-1 WT (full bars) or *NLRP3^−/−^* (open bars) cells after 24 h treatment with 10 ng/mL LPS. Mean ± SD of *N* = 5, *** *p* < 0.001 compared to the respective control, ^###^
*p* < 0.001 compared to WT cells.

**Figure 4 biomolecules-12-01095-f004:**
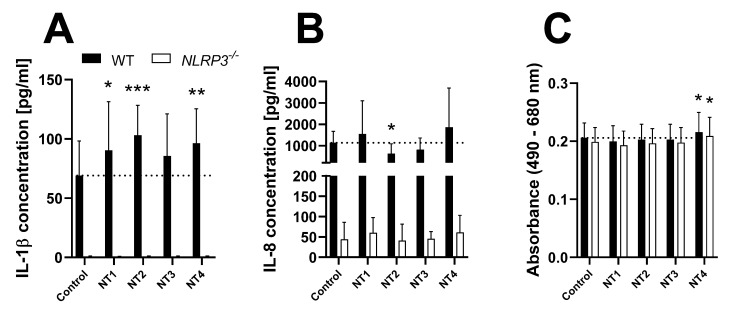
Release of IL-1β (**A**) and IL-8 (**B**), as well as LDH activity (**C**) in the supernatant of PMA-differentiated THP-1 WT (full bars) or *NLRP3^−/−^* (open bars) cells after 24 h treatment with 50 µg/cm^2^ of four different TiO_2_ samples. Cytokine data is depicted as relative values compared to the WT control, LDH data is depicted as relative values compared to the respective control. Mean ± SD of *N* = 4, * *p* < 0.05, ** *p* < 0.01, and *** *p* < 0.001 compared to the respective control.

**Figure 5 biomolecules-12-01095-f005:**
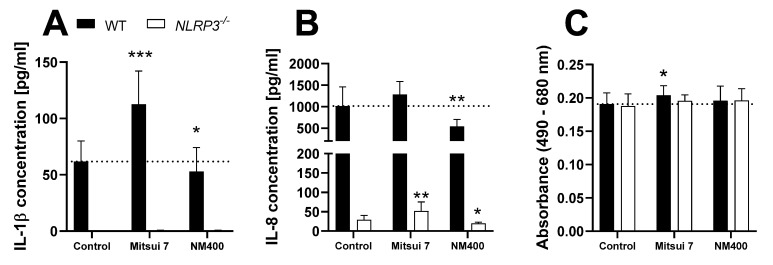
Release of IL-1β (**A**) and IL-8 (**B**), as well as LDH activity (**C**) in the supernatant of PMA-differentiated THP-1 WT (full bars) or *NLRP3^−/−^* (open bars) cells after 24 h treatment with 50 µg/cm^2^ of two different MWCNTs. Cytokine data is depicted as relative values compared to the WT control, LDH data is depicted as relative values compared to the respective control. Mean ± SD of *N* = 4, * *p* < 0.05, ** *p* < 0.01, and *** *p* < 0.001 compared to the respective control.

**Figure 6 biomolecules-12-01095-f006:**
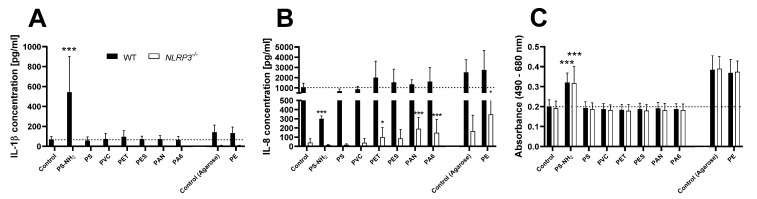
Release of IL-1β (**A**) and IL-8 (**B**), as well as LDH activity (**C**) in the supernatant of PMA-differentiated THP-1 WT (full bars) or *NLRP3^−/−^* (open bars) cells after 24 h treatment with 50 µg/cm^2^ of different plastic particles. Cytokine data is depicted as relative values compared to the WT control, LDH data is depicted as relative values compared to the respective control. Mean ± SD of *N* = 4, * *p* < 0.05 and *** *p* < 0.001 compared to the respective control.

**Table 1 biomolecules-12-01095-t001:** TiO_2_ samples, carbon nanotubes, and MNPs used for experiments.

	Material	Abbreviation	Size	Source	Characterization/ Previous Use
**Titanium dioxides**	TiO_2_ (P25)	NT1	12–18 nm	Evonik, Essen, Germany	[29]
	TiO_2_ (PC105)	NT2	10 nm	Cristal, Jeddah, Saudi Arabia	[29]
	TiO_2_ (SX001)	NT3	12–15 nm	Solaronix, Aubonne, Switzerland	[29]
	TiO_2_ (UT001)	NT4	16–17 nm	UNITO, Turin, Italy	[29]
**Carbon nanotubes**	Multi-walled carbon nanotubes 1 “Mitsui”	Mitsui 7	40–100 nm (diameter); 13 µm (length)	Mitsui & Co., Tokyo, Japan	[31]
	Multi-walled carbon nanotubes 2 “NM400”	NM400	30 nm (diameter); 5 µm (length)	JRC repository, Ispra, Italy	[31]
**Micro/nanoplastics**	Amine-modified polystyrene spheres	PS-NH_2_	50 nm	Sigma Aldrich, Schnelldorf, Germany (L0780)	[21]
	Polystyrene spheres	PS	50 nm	Polysciences Inc., Warrington, PA, USA (08691–10)	[21]
	Polyvinyl chloride powder	PVC	235 nm (mode)	Werth–Metall, Erfurt, Germany	[21]
	Polyethylene spheres	PE	611 nm (mode)	Cospheric LLC, Santa Barbara, CA, USA	[36]
	Polyethylene terephthalate fragments	PET	16 nm (nanofraction); 5.7 µm (microfraction)	Produced via laser ablation	
	Polyester fibers	PES	17.5 µm (diameter); 10 µm (length)	Produced via cryotome	
	Polyacrylonitrile fibers	PAN	18.5 µm (diameter); 10 µm (length)	Produced via cryotome	
	Polyamide 6 (nylon) fibers	PA6	27.5 µm (diameter); 10 µm (length)	Produced via cryotome	

## Data Availability

Not applicable.

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
