# Peer review of "Assessing the NLRP3 Inflammasome Activating Potential of a Large Panel of Micro- and Nanoplastics in THP-1 Cells"

_biomolecules, 2022, doi:10.3390/biom12081095_

Round 1
Reviewer 1 Report
The manuscript biomolecules-1789230 by Busch et al. describes the development and application of genetically modified THP-1 cells (NLRP3 knockout), and control WT cells, to the assessment of the NLRP3 activation potential of micro and nanoplastics (MNPs). The work is well written, of general interest, and aligned with the 3R principle for the use of animals in research. I have, however, a few concerns that, in my opinion, should be addressed before publication:
- I am not convinced the authors can say they “validate” the model just by comparing the results they obtained using the developed cell line with the ones obtained by others in independent works. I find the word “comparison” much more applicable in this context.
- Maybe I am old fashioned, but in my opinion the manuscript does not prove that the cells are not expressing NLRP3. This is particularly relevant, again, when claiming validation. Please, provide western blot (e.g.) analysis of the cells.
- Because the cells are GFP-transfected, it would also be interesting to show some fluorescence images of the cells. It would make the work more attractive and convincing.
- How do the authors account for natural differences in the number of cells between experiments in the LDH assay? Usually, the LDH in the supernatant is corrected for the “total LDH” value, which correlates with the number of cells in each well. If this correction was performed, it is not clear from the text, and maybe the equation that was used for the calculation of the LDH activity should be included, as well, in the manuscript.
- Note that the definition of the decimals and thousands in numbers needs to be homogenized throughout the manuscript: in section 2.2 we see “18,2 MW/cm” and “1.064 nm”, while in the rest of the manuscript the “.” is used for decimals while the thousands have no punctuation. Please, be consistent.
- Considering that the experiments to assess the interference of the MNPs on the ELISA and LDH assays showed some interference for a few parameters, why was the experiment not repeated? Couldn’t this be just a random finding? Also, in case the interference is confirmed, it is important that the authors convey how they deal with that interference. For example, PS-NH2 seem to somehow increase the signal from IL8 (Fig S7B), but in Fig 6B the tendency shown is a decrease. How do you explain this? Is the interference already accounted for in this result? If the interference is not accounted for in Fig 6B, then I suggest simply removing the result from the graph in order not to be misleading.
- In my understanding, the sentence in lines 351-353 is not correct, according to the results of Fig6B.
- In line 174 it seems a reagent is missing, referring to “4 mg/mL”.
- Please, explain what you consider to be “biological replicates”, in line 215.
- Looking at Fig S2, the description included in lines 229-230 does not seem obvious… are the morphologies so similar?
- Please, check the wavelength mentioned for the H-C=O groups in line 243
Reviewer 2 Report
In the manuscript biomolecules-1789230, Busch and coworkers study the NLRP3 inflammasome activation by a variety of micro and nano plastics. The topic is clearly of interest, the authors have produced and characterized an array of different and representative plastics, and the idea of verifying the specificity of the NLRP3 inflammasome by genetic inactivation of the NLRP3 gene was clearly an excellent scientific choice. However and as usual, the devil hides in the details and there are several points that appear at least questionable for me. I will therefore start with these major scientific issues
Major comments
1) From the experimental section as it is written, I do not understand how the cells were really exposed to the plastics (lines 169-188). After the differentiation step with PMA, cells were resuspended in medium (line 172). The same complete medium than in lines 150-152 or a different one ?
Furthermore, what happened after the 1 hour reattachment period. Were the plastics suspensions added to the 1ml of initial (unknown) medium and if yes, which volume ? Was the reattachment medium removed and replaced by the plastics suspensions in treatment medium ? If this is the case, why has the 1% FCS concentration been selected in the treatment medium ? The only recent paper where I could find a reduced serum concentration is the following (doi: 10.3389/ftox.2021.780778) but when reading it the rationale for reducing the serum concentration was to reduce proliferation, which does not hold with PMA-differentiated THP1 cells, which do not proliferate after the PMA treatment. So the rationale of this unusual FCS concentration must be detailed and if it comes from any reference, this work should be cited. Indeed, compared to the in vivo situation where the cells bathe in a liquid containing tens of mg of proteins per ml, even 10% serum is a low approximate (only ca. 5mg/ml) and 1% FCS is even worse.
In the same trend, the final volumes of treatment medium used in the experiments should be detailed.
2) The authors have used titanium dioxide nanoparticles as a positive control, and refer to a previous paper of theirs (ref 29). When reading this paper (and others) it appears that the classical (and correct) positive control for inflammasome activation in silica, not titanium dioxide. Is there any solid scientific reason for having selected a less efficient positive control (as can be seen in 29) than the classical one?This gives the unpleasant impression of either self-citation or decreasing the values of the positive control to make the values of the tested substances look bigger. In any case, this is difficultly acceptable.
3) The statistics used in the present manuscript appear questionable, to say the least.
Quoting lines 214-215 : In vitro experiments were performed in at least four independent runs with two biological replicates. This is completely unclear to me. Does this mean that two biological replicates were used with each biological point seeded in two wells each? Apart from a much needed clarification, two biological replicates are not enough to derive any solid conclusion.
Quoting lines 217-219: Statistical analysis was performed 217 with two-way ANOVA and Dunnett’s multiple comparison test. A p-value of <0.05 was 218 considered statistically significant.
As the two tests are quite different in their purposes and power, which p value is selected for declaring a result significant ? The Dunnett’s one? Furthermore, while I see the interest of a Dunnett’s test, I do not see the interest of an ANOVA when the purpose is to compare several plastics samples to the negative control.
In the same trend, the change of data displaying options between figure 3 and the following ones appears strange and disturbing, to say the least. I just want to see the real pg/ml figures, not ratios. This is clearly part of a minimal data availability policy, and I do not agree with the authors in their claim that data availability statement is not applicable (line 454). I must confess that I am somewhat astonished by the fact that 1.8±1 is significantly different from 1 (Fig. 6B PET bar), as I am for Fig 5B, for the Mitsui 7 nanotubes.
4) At no point in the discussion do the authors envision the possibility that the biological reactions induced by the PET particles may come from a preparation artifact, while there are quite a few indices thereof. The first one is the transformation of a transparent material into a white one (lines 114-115), which indicates a transformation of the structure of the material. Furthermore, laser ablation implies high energy transfer to the material, which may well result in surface chemistry modifications. For example, fracturation of silica has been shown to alter the surface chemistry, induce oxidative potential and strong biological responses (doi: 10.1186/s12989-016-0136-6; doi:10.1073/pnas.2008006117). Thus, their results on PET particles should be taken with great caution.
Minor
Crytome does not exist, cryotome does.
All in all, serious reservations can be cast on the data presented in this paper and their documentation, so that the manuscript cannot be considered any further prior to addressing all of these. Furthermore, once proper documentation will have been given and the correct positive control added, the data may still prove unsuitable for publication. The change in the positive control will certainly not disqualify the experimental data, but the discussion may prove completely different. Showing a pro-inflammatory effect (IL-8 release) that is close the one of the so-called positive control here but may be one tenth of the one of the real positive control is clearly not the same story and cannot be discussed in the same way.
In addition, the data may prove unsuitable because of insufficient statistical power or irrelevant cell culture choices. Only a revised version will tell.
Reviewer 3 Report
In this manuscript, the authors describe the inflammatory effects of microplastics on immune system cells. I particularly appreciate this work, I find it complete in every part. It only needs some formatting adjustments
Page 3 is practically blank, it would be better to fix this small layout problem
Figures 1-2 add standard deviation, I guess the production of microplastics has been done more than once, and the same number of particles of a specific size will not always have been obtained
Round 2
Reviewer 1 Report
Following the reviewer’s comments, I find that the authors have only made minor changes to the original manuscript. Most of the points raised by the reviewers might have been discussed in the authors’ reply, but did not produce any practical differences in the manuscript. As an example, the authors provide Figure 1 in their reply, but this issue is not mentioned at all in the manuscript. I remind the authors that the issues raised by the reviewers could translate concerns from (future) readers.
The authors’ reply to my comment #6, for example, does not really answer my question whether the interference was accounted for or not in the displayed result. I understand that the decrease in IL8 is not significant (that’s why I referred to it as a “tendency” rather than a “difference”) but this does not exclude the need to openly disclose how the results were calculated. I would make a similar comment to the LDH calculation: if your results were obtained in a different way than the conventional one, this should be clearly described in the manuscript text.
I still think the work is interesting, but I am not convinced it is ready to be published.
Reviewer 2 Report
The revision of the manuscript biomolecules-1789230 has been minimal and has addressed only the more technical concerns, i.e. the exposure protocol and the comment on the PET particles. By the way there is in the literature a protocol preparing PET nanoparticles by a wet route (Rodriguez-Hernandez et al DOI: 10.1039/c9en00365g). Even though it uses strong acidic media with a foreseeable degradation of the PET polymer, it should not give the cycle hydroxylation observed with laser ablation.
However, the two major concerns, i.e. the choice of serum concentration, the nature of the positive control and the cytokines induction, have not been properly addressed.
Regarding the choice serum concentration, converting the authors’ response in plain language would give something as: « because this bloody (sic) serum has anti oxidative properties that would decrease the effects that I want to see and because I want to see effects, I will reduce the serum concentration down to the point where the cells are still viable ». This is not only non-physiological, it is anti-physiological in the sense that the cells in vivo bathe in pure serum and operate at low oxygen tension. Classical cell culture operates at high oxygen tension and at a reduced serum proportion (10%), reducing further the serum concentration goes against the physiology. This reduction may be argued for on a case by case basis on a physiologically-grounded discussion, but not on a convenience argument as in the present case. The papers cited by the authors do not bring any strong argument in favor of this serum reduction, and even though it should be argued in the context of macrophage physiology and plastic exposure, something that is completely missing. This alone argues for manuscript rejection.
Regarding the cytokinic response of THP-1, the authors reply has in fact validated the fears that I had on the initial version of the manuscript. The bare truth is that the basal response of THP-1 cells is highly variable from one experiment to another, so that experiments are difficult to compare and only indirect indexes (fold-changes) can be used. Even though, huge standard deviations can be seen (e.g. Figures 4 and 6), which further substantiates the high variability of this system. In this context, the claim on the quality of the THP-1 system made in lines 385-387 appears non-grounded, to say the least.
Regarding the statistical techniques themselves, I do not think that parametric methods such as ANOVA and Dunnett’s tests are valid, as the normal distribution cannot be guaranteed. The authors should have used a Kruskal-Wallis test in place of the ANOVA and Mann-Whitney as a post-hoc test. As the number of conditions tested is still relatively small, the multiple testing type I error will be moderate. The point is that these non parametric tests are often less powerful than parametric ones and thus require data of higher quality to obtain conclusions, which may prove risky with the data available here. It is true that statistics are made to reveal the predictive power of the data, but the diversity of statistical methods and of data transformation is such that over-interpretation is a major caveat. As the classical joke on statistics says, to the question ‘’ how much is 2+2 ?’’ the mathematician replies 4 and the statistician replies ‘’how much do you want it to be ? ’’
Finally regarding the positive control issue, the point risen by the authors is clearly not valid either. By making a comparison, it is exactly as if authors would use LTA instead of LPS as a positive control for inflammation induction just because in one papers of theirs that have performed this comparison. It is clearly not a problem of documenting minor effects, which I completely agree is definitely necessary, it is a problem of comparing them to the adequate and classical positive control. As a reviewer, the publication strategy of the authors is not my problem. To conclude on the necessity of documenting minor effects, the Dostert et al. paper cited by the authors in their response does not argue for the necessity of documenting minor effects in vitro, which I repeat is completely necessary, it rather further argues in the favor of the poor performance of the THP-1 system.
All in all the authors have not addressed by major concerns in this revision.
